# Grey-Fuzzy Hybrid Optimization for Thermohydrodynamic Performance Prediction of Misaligned Rough Elliptic Bore Journal Bearing

**Sushanta Kumar Pradhan** [1], **Ramanuj Kumar** [2] **and Prakash Chandra Mishra** [1,*]

1 Department of Mechanical Engineering, Veer Surendra Sai University of Technology, Burla 768018, India
2 School of Mechanical Engineering, KIIT University, Bhubaneswar 751024, India
* Correspondence: pcmishra_me@vssut.ac.in; Tel.: +91-891-7535445

**Abstract:** Rough elliptic bore misaligned journal bearing performance involves many geometric and operational parameters, which directly or indirectly affect the thermohydrodynamic performance output. Improper design and manufacturing of journal bearings lead to enhanced friction, reduced operational life, and poor serviceability. A rule of thumb is to understand the operational efficiency of the bearing through modelling and simulation and to implement the knowledge of bearing critical parameters in manufacturing and operation. Therefore, decision-making in bearing parameter selection is a crucial process, for which several optimization tools and techniques have been developed from time to time. Moreover, these techniques have their own merits and demerits. This paper proposes a grey-based fuzzy approach to optimize the thermohydrodynamic performance of journal bearings with roughness, bore non-circularity, and shaft misalignment. Based on the results, the optimal level of factors is $\varepsilon_1$ (0.3)-$\beta_1$ (0.5)-$G_3$ (2)-$y_1$ (0.1), while at this condition, the optimal solutions for responses, such as $W_{is}$, $W_{th}$, $F_{is}$, $F_{th}$, $Q_{is}$, and $Q_{th}$ are 3.684, 2.84, 165.2, 178.3, 5.67, and 6.32, respectively.

**Keywords:** elliptic bore journal bearing; statistical properties; grey-fuzzy algorithm; misalignment

## 1. Introduction

Journal bearings with an elliptic bore are an all-time possibility in machine application due to manufacturing constraints and small-amplitude vibrations during machining operations. With the wide-scale application of journal bearings, it is necessary to develop an appropriate methodology to understand the effect of small-scale irregularities, such as roughness and large-scale irregularities, such as bore-out-of-roundness on the lubrication performance.

In 1966, Hamilton et al. [1] developed a micro-scale lubrication model to analyze the effect of surface irregularities on hydrodynamic pressure development and the load carrying capacity of a contact conjunction. After a period of time, many numerical and experimental investigations were carried out from time to time into the mechanism of load carrying capacity (LCC). The LCC is an output that depends on many geometrical and operational parameters. The optimization of journal bearing performance using an enhanced artificial neural network [2] (ANN) is one commendable contribution, which is an intelligence system developed to minimize film temperature and lubricant flow in a short journal bearing. A comparative analysis of genetic algorithms and successive quadratic programming shows good agreement and reliability of ANN in optimizing bearing problems.

Moreover, in regard to bearing optimization, Song et al. [3] developed the EALA algorithm. It is a hybrid model that deals with an artificial life algorithm (ALA) and a random tabu method (r-tabu method) to eliminate the demerits of the individual methods. Furthermore, Hirani et al. [4] developed an optimization technique to minimize the lubricant flow-in and power loss of a journal bearing subjected to a steady load. This

design method uses radial clearance, L/D ratio, oil groove geometry, oil viscosity, and supply pressure as input variables. The uncertainties that affect the load carrying capacity can be analyzed using fuzzy logic [5]. The stochastic Monte Carlo simulation is useful in uncertainties of both the design and maintenance of tilt pad bearing. Diwedi [6] predicted a fuzzy approach for the selection of fluid film bearing (hydrostatic, hydrodynamic or hybrid journal bearing). Decision-making with fuzzy logic is found in the case of magnetic bearing for stability analysis [7], using the linear matrix inequality method to maximize stability [8] and performance. Moreover, this process has been proven to be helpful in the critical analysis of design, manufacturing, and operation of various systems, including journal bearing, rolling element bearing [9,10], and manufacturing process, such as hard turning [11,12].

Bearing design charts are often based on a smooth circular bore with aligned shaft considerations. Therefore, the predicted performance through charts differs from the experimental performance output. To converge the numerical and experimental performance output, more accurate dimensions of bearing systems need to be quantified, including bore irregularities and shaft misalignment. Although the current state-of-the-art for journal bearing simulation and modelling [13] addressed bore ellipticity [14,15] non-circularity, stochastic/deterministic roughness pattern [16,17] misalignment [18] thermos-hydrodynamic [17,19], cavitation-induced flow in bearing [20], etc., individually or in combination, the involvement of multiple parameters in determining the rough elliptic bore misaligned journal bearing performance as well as the critical analysis of these parameters and their highly non-linear effect on bearing performance should be investigated through the application of prediction tools, such as the grey-fuzzy hybrid optimization technique. Therefore, this is the main contribution of this research paper.

## 2. Theories of the Model

### 2.1. Film Thickness of Misaligned Rough Elliptic Bore Journal Bearing

Film thickness is one of the primary parameters for bearing performance evaluation. Mishra [18] developed a film parameter for rough elliptic bore misaligned journal bearing, involving bore eccentricity($\varepsilon$), ellipticity ($G$), misalignment ($D_m$), transverse, and longitudinal roughness [21] orientation ($h_t$, $h_l$). The instantaneous film profile of rough elliptic bore misaligned journal bearing in [16] is shown in Equation (1):

$$h_s = c \left\{ \begin{array}{l} 1 + G\cos^2(\theta - \alpha) + \in_o \cos(\theta - \phi_o) \\ + \psi\tau^* \cos(\theta - \alpha - \phi_o) \end{array} \right\} - h_t + h_l \tag{1}$$

where

$$\psi = D_m \psi_{\max}$$

In addition,

$$\psi_{\max} = 2 \left\{ \left(1 - \in_o^2 \sin^2 \alpha \right)^{\frac{1}{2}} - \in_o \cos \alpha \right\}$$

The model has a limitation of working within $0.1 < \varepsilon < 0.9$, $0 < D_m < 0.9$, $0 < G < 3.0$. The roughness pattern in both transverse and longitudinal directions is assumed to be a sinusoidal wave with a roughness amplitude of 5 µm. The schematic of the misaligned rough elliptic bore journal bearing is shown in Figure 1. Table 1 shows the expectancy operators for the rough bearing.

$$h_t = A_t \sin\left(\frac{2\pi x}{\lambda t}\right)$$

In addition,

$$h_l = A_l \sin\left(\frac{2\pi x}{\lambda l}\right)$$

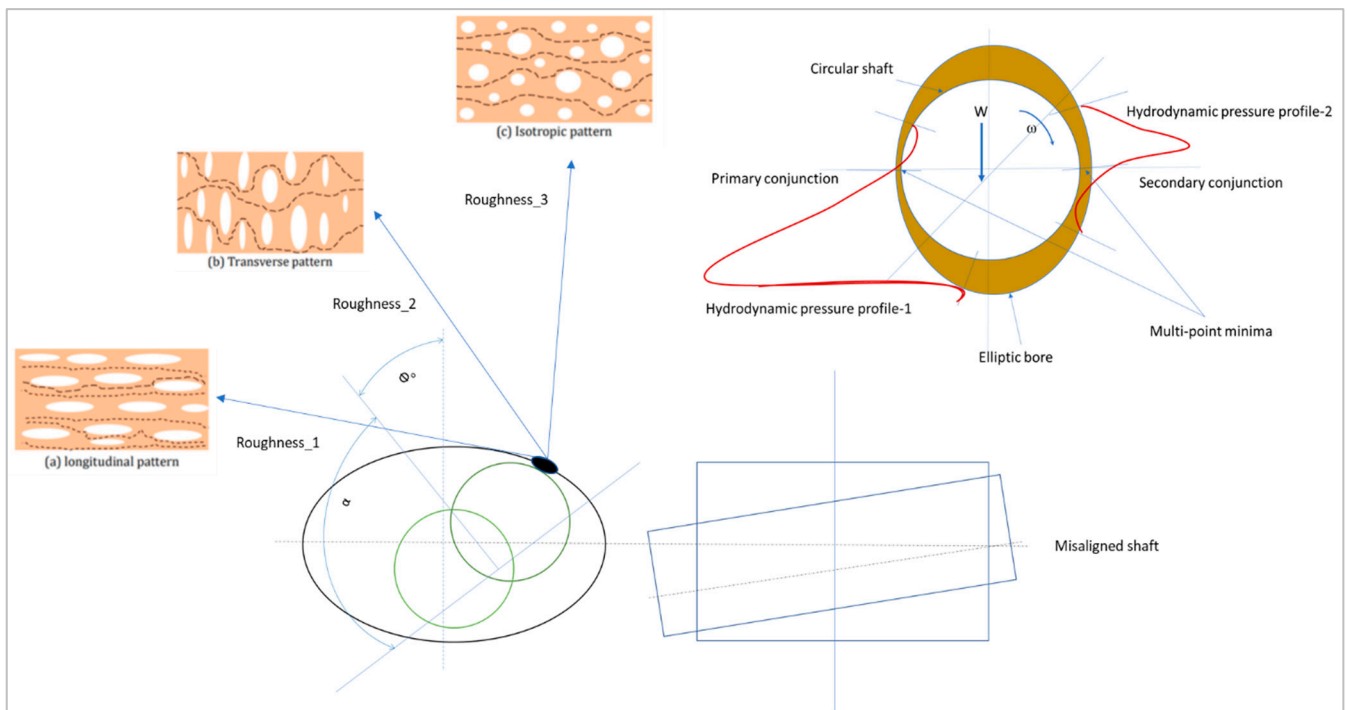

**Figure 1.** Misaligned rough elliptic bore journal bearing schematic.

**Table 1.** Expectancy operator.

| Film Terms | Value |
|:---:|:---:|
| $\tilde{\xi}(h)$ | $1 + G \cos^2 \theta + \in \cos \theta$ |
| $\xi\left(\frac{1}{h}\right)$ | $\frac{35}{37y^7}\left\{6\left(y^2 - h^2\right)\log\left(\frac{y+h}{y-h}\right) + \frac{2}{5}yh\left(15h^4 - 40y^2h^2 + 33y^4\right)\right\}$ |
| $\xi\left(h^3\right)$ | $h^3 + \frac{hy^2}{3}$ |

## 2.2. Thermohydrodynamics of Misaligned Rough Elliptic Bore Journal Bearing

Lubrication performance is quantitatively determined using the Reynolds equation [18]. Both hydrodynamic as well as elasto-hydrodynamic lubrication principles were developed mathematically to formulate and solve the film pressure by involving the film profile, lubricant rheology, shaft rotational speed, etc., as the input to these models [18]. The Reynolds equation is case-specific and involves modification based on geometry and operational requirements. Dien et al. [22] developed a time-dependent Careau viscosity model based on the modified Reynolds equation, as shown in Equation (2).

$$\frac{\partial}{\partial x}\left(\frac{\rho\xi\left(h^3\right)}{\eta}\frac{\partial p}{\partial x}\right) + \frac{\partial}{\partial z}\left(\frac{\rho\xi\left(h^3\right)}{\mu^*}\frac{\partial p}{\partial z}\right) = 6\omega r_j\left(1 - \frac{2}{\omega}\overset{*}{\Phi}\right) + 12\rho\overset{*}{e}\cos(\theta - \Phi) \quad (2)$$

where

$$\lambda = \mu^* + 2\lambda^2\left(\frac{u_2 - u_1}{\xi(h)}\right)^2(\mu_0 - \mu_\infty)\left(\frac{n-1}{2}\right)\left(1 + \lambda^2\left(\frac{u_2 - u_1}{\xi(h)}\right)^2\right)^{\frac{n-3}{2}}$$

In addition,

$$\mu^* = \mu_\infty + (\mu_\infty - \mu_0)(1 + \lambda^2\Gamma)^{\frac{n-3}{2}}$$

Moreover,

$$\Gamma = \frac{(u_2 - u_1)^2 + (v_2 - v_1)^2}{\xi(h^2)}$$

Jamali and Al-Hammad [20] developed a dimensionless density ($\theta_c = \rho/\rho_{cav}$) based on the Reynolds equation, which is capable of performance evaluation in both the full-fluid film and cavitation region [16], as shown in Equation (3).

$$\frac{\partial}{\partial x}\left(\frac{\rho_c\xi(h^3)g\beta_b}{\eta}\frac{\partial\theta_c}{\partial x}\right) + \frac{\partial}{\partial z}\left(\frac{\rho_c\xi(h^3)g\beta_b}{\mu^*}\frac{\partial\theta_c}{\partial z}\right)$$
$$=$$
$$6\omega r_j\left(1 - \frac{2}{\omega}\overset{*}{\Phi}\right)\frac{\partial\rho_c\xi(h)\theta_c}{\partial x} + 12\rho_c\theta_c\overset{*}{e}\cos(\theta - \Phi) \tag{3}$$

The finite difference method is used to solve these second-order partial differential equations. The boundary value problem involves mass-conserving boundary conditions. The method involves a switch function, $g(\theta_c)$, that plays an importance role in the cavitation index, where $g(\theta_c) = 1$ for $p \geq p_{cav}$ and $\rho \geq \rho_{cav}$, while $g(\theta_c) = 0$ for $p \leq p_{cav}$ and $\rho \leq \rho_{cav}$. In this Reynolds equation, the Poiseuille term is centrally discretized, while the Couette equation uses backward differentiation. The latter is more dominant in the cavitation region. The numerical technique involves the low-relaxation effective influence Newton-Raphson iterative method for pressure error convergence. In the zone of cavitation, the liquid mass content is controlled by $\rho_{cav}$, $\theta_c$, and $h$ (as $\rho_{cav}.\theta_c.h$), where h is the film thickness. This variable is the dimensionless density, which is the fraction of lubricant content in the cavitation zone. Here, $(1 - \theta_c)$ is the cavity void fraction. When a solution to $\theta_c$ is obtained, the pressure profile can be reconstructed. The unwrapped bearing surface is discretized to form a grid of ($94 \times 16$), which gives a square grid element for the stable hydrodynamic pressure [18] of film geometry. Film geometry plays an important role in the 3D hydrodynamic pressure profile. Misalignment reduces the longitudinal distribution of the pressure into a small area of bearing surface with a more concentrated pressure profile [18]. Due to non-circularity in terms of ellipticity, secondary pressure bumps with reduced peaks are formed. The roughness-induced pressure profile distorted from the smooth profile is due to many micro-scale lubricating conjunctions.

Rheology of the model. The hydrodynamic pressure in a contact conjunction was greatly affected due to an increase in the temperature of lubricant as a result of shear thinning. This thermally modified pressure can be estimated through viscosity-pressure-temperature interrelationship and density-pressure-temperature interrelationship, as shown by Roeland [23] and Dowson-Higginson [24]

$$\mu^* = \mu_0 \exp(\ln \mu_0 + 9.67)\left(-1 + \left(1 + 5.1 \times 10^{-9}p\right)^z\right) - \gamma(T - T_0) \tag{3a}$$

In addition,

$$\rho = \rho_0\left(1 + \frac{0.6 \times 10^{-9}p}{1 + 1.7 \times 10^{-9}p}\right)(1 - \beta_t(T - T_0)) \tag{3b}$$

### 2.3. Numerical Method for Film Temperature Evaluation

The intermediate lubricant layer moves rapidly and relatively, due to which fluid friction arises. This friction leads to a rise in temperature due to frictional heat generation [25]. The temperature distribution across the film depends largely on the fluid flow and rapid change in fluid geometry. Heat generation in the lubricating system is due to the heat of viscous shear and compression, whereas heat generation out of the system is due to cooling by convection across the oil film and by conduction through the shaft and sleeve. In Equation (4), the energy equation using the Carreau viscosity model [18] is shown, which is a partial differential equation involving film pressure and temperature as variables. It

can be solved by the finite difference method, with a known initial value of pressure and a boundary value of temperature.

$$\left\{ \left( \rho c_f \frac{u_2}{2}\xi(h) - \rho c_f \frac{\xi(h^3)}{12\eta}\frac{\partial p}{\partial x} \right)\frac{\partial T_m}{\partial x} - \left( \rho c_f \frac{\xi(h^3)}{12\mu^*}\frac{\partial p}{\partial z} \right)\frac{\partial T_m}{\partial z} \right\}$$

$$=$$

$$\left\{ \begin{array}{c} \mu^*\left( \frac{u_2{}^2}{\xi(h)} \right) + \left( \beta_t\xi(h)T_m\frac{u_2}{2} \right)\frac{\partial p}{\partial x} + \left( \frac{\xi(h^3)}{12\eta} - \beta_t T_m\frac{\xi(h^3)}{12\eta} \right)\left( \frac{\partial p}{\partial x} \right)^2 \\ + \left( \frac{\xi(h^3)}{12\mu^*} - \beta_t T_m\frac{\xi(h^3)}{12\mu^*} \right)\left( \frac{\partial p}{\partial z} \right)^2 + \beta_t\xi(h)T_m\frac{\partial p}{\partial T} - \rho c_f\xi(h)\frac{\partial T_m}{\partial t} \end{array} \right\} \tag{4}$$

The mean temperature is derived from Equation (4) and shown as Equation (5).

$$T_m = \frac{\left\{ \begin{array}{c} \left( \rho c_f\frac{u_2}{2}\xi(h) - \rho c_f\frac{\xi(h^3)}{12\eta}\frac{\partial p}{\partial x} \right)\frac{\partial T_m}{\partial x} - \left( \rho c_f\frac{\xi(h^3)}{12\mu^*}\frac{\partial p}{\partial z} \right)\frac{\partial T_m}{\partial z} - \\ \mu^*\left( \frac{u_2{}^2}{\xi(h)} \right) - \frac{\xi(h^3)}{12\eta}\left( \frac{\partial p}{\partial x} \right)^2 - \frac{\xi(h^3)}{12\mu^*}\left( \frac{\partial p}{\partial z} \right)^2 - \rho c_f\xi(h)\left( \frac{\partial T_m}{\partial t} \right) \end{array} \right\}}{\left( \beta_t\xi(h)T_m\frac{u_2}{2} \right)\frac{\partial p}{\partial x} - \beta_t\frac{\xi(h^3)}{12\eta}\left( \frac{\partial p}{\partial x} \right)^2 - \beta_t\frac{\xi(h^3)}{12\mu^*}\left( \frac{\partial p}{\partial z} \right)^2 + \beta_t\left( \frac{\partial p}{\partial t} \right)} \tag{5}$$

The solution of the abovementioned equation of temperature uses the finite difference method to discretize the right-hand side with the central differentiation technique [26]. The low-relaxation iterative algorithm is used for temperature error convergence. The generated conjunction heat is dissipated to the rotating journal and stationary sleeve. It is assumed that the heat of convection through the fluid film in the flow direction (along circumference) is significantly less compared to the side leakage direction (along the bearing length). The temperature at the first node is considered ambient, which indicates that $T(0, z) = T_o$. The temperature at the inlet is equal to the temperature at the outlet along the length of the bearing, which indicates that $T(\theta, z) = T(\theta, -z)$. Table 2 shows the model input parameters required for the simulation. The bearing considered for analysis is a self-aligning journal bearing with bearing code of SA 35M/S4170, which is lubricated with an oil ring and has a maximum load capacity of 10 kN.

**Table 2.** Model input parameters.

| Input Variables | Values | Input Variables | Values |
|---|---|---|---|
| Shaft radius, $(r_j)$ | 0.025 m | Characteristic relaxation | $4.0 \times 10^{-6}$ |
| Bearing length, (L) | 0.030 m | Time constant, $(\lambda)$ | $3.0 \times 10^{-6}$ |
| Radial clearance, (c) | 60 µm | Lubricant density, $(\rho)$ | 860 kg/m$^3$ |
| Rotational speed, (N) | 7000 rpm | Lubricant specific heat, $(C_f)$ | 1950 J/kgK |
| Wavelength of roughness, $(\gamma_l/\gamma_t)$ | 0.005 m | Inlet lubricant temperature, $(T_o)$ | 315 K |
| Lubricant details | | Viscosity pressure index, (Z) | 0.65 |
| Lubricant type | SAE10W50 | Temperature coefficient of viscosity, $(\gamma)$ | 0.0310225 K$^{-1}$ |
| Limiting viscosity at low shear rate $(\mu_0)$ | 0.105 PaS | Coefficient of thermal expansion, $(\beta_t)$ | 0.000792 K$^{-1}$ |
| Limiting viscosity at high shear rate $(\mu_\infty)$ | 0.055 PaS | | |

The load carrying capacity of the bearing is estimated by integrating the hydrodynamic pressure over the area of conjunction. The resolved component of this reactive force is shown as:

$$F_x = -\int_0^L \int_0^\theta (pr\cos\theta)d\theta dz \tag{6}$$

$$F_y = - \int\limits_0^L \int\limits_0^\theta (pr \sin \theta) d\theta dz \tag{7}$$

Furthermore, the hydrodynamic action can be resolved in radial and circumferential directions as follows:

$$F_r = -F_x \cos \phi - F_z \sin \phi \tag{8}$$

$$F_\phi = -F_x \sin \phi - F_z \cos \phi \tag{9}$$

The resultant bearing load and attitude angle is shown as:

$$W = \left(F_r^2 + F_\phi^2\right)^{\frac{1}{2}} \tag{10}$$

$$\phi = \tan^{-1}\left(-\frac{F_\phi}{F_r}\right) \tag{11}$$

Linearization of bearing equation [27] is shown as:

$$\frac{1}{r_j^2} \frac{\partial}{\partial \theta} \left( \frac{\rho \xi (h_s{}^3)}{12\eta} \frac{\partial p_s}{\partial \theta} \right) + \frac{\partial}{\partial z} \left( \frac{\rho \xi (h_s{}^3)}{12\mu^*} \frac{\partial p_s}{\partial z} \right) = \frac{U}{2r_j} \frac{\partial \rho \xi (h_s)}{\partial \theta} \tag{12}$$

Or

$$\begin{aligned}
&\frac{1}{r_j^2} \frac{\partial}{\partial \theta} \left( \frac{\rho \xi (h_s{}^3)}{12\eta} \frac{\partial p_x}{\partial \theta} \right) + \frac{\partial}{\partial z} \left( \frac{\rho \xi (h_s{}^3)}{12\mu^*} \frac{\partial p_x}{\partial z} \right) \\
&= \\
&\frac{U}{2r_j} \frac{\partial (\rho \cos \theta)}{\partial \theta} - \frac{1}{r_j^2} \frac{\partial}{\partial \theta} \left( \frac{\rho \xi (h_s{}^2) \cos \theta}{4\eta} \frac{\partial p_s}{\partial \theta} \right) \\
&- \frac{\partial}{\partial z} \left( \frac{\rho \xi (h_s{}^2) \cos \theta}{4\mu^*} \frac{\partial p_s}{\partial z} \right)
\end{aligned} \tag{13}$$

Therefore,

$$\frac{1}{r_j^2} \frac{\partial}{\partial \theta} \left( \frac{\rho \xi (h_s{}^3)}{12\eta} \frac{\partial p_{x*}}{\partial \theta} \right) + \frac{\partial}{\partial z} \left( \frac{\rho \xi (h_s{}^3)}{12\mu^*} \frac{\partial p_{x*}}{\partial z} \right) = \rho \cos \theta \tag{14}$$

In addition,

$$\frac{1}{r_j^2} \frac{\partial}{\partial \theta} \left( \frac{\rho \xi (h_s{}^3)}{12\eta} \frac{\partial p_{z*}}{\partial \theta} \right) + \frac{\partial}{\partial z} \left( \frac{\rho \xi (h_s{}^3)}{12\mu^*} \frac{\partial p_{z*}}{\partial z} \right) = \rho \sin \theta \tag{15}$$

The dimensionless spring and damping coefficients are shown as:

$$\begin{pmatrix} K_{xx} & K_{xz} \\ K_{zx} & K_{zz} \end{pmatrix} = -\frac{c}{W} \begin{pmatrix} \int\limits_0^L \int\limits_0^{2\pi} p_x r_j \cos \theta d\theta dz & \int\limits_0^L \int\limits_0^{2\pi} p_z r_j \cos \theta d\theta dz \\ \int\limits_0^L \int\limits_0^{2\pi} p_x r_j \sin \theta d\theta dz & \int\limits_0^L \int\limits_0^{2\pi} p_z r_j \sin \theta d\theta dz \end{pmatrix} \tag{16}$$

$$\begin{pmatrix} B_{xx} & B_{xz} \\ B_{zx} & B_{zz} \end{pmatrix} = -\frac{c\omega}{W} \begin{pmatrix} \int\limits_0^L \int\limits_0^{2\pi} p_{x*} r_j \cos \theta d\theta dz & \int\limits_0^L \int\limits_0^{2\pi} p_{y*} r_j \cos \theta d\theta dz \\ \int\limits_0^L \int\limits_0^{2\pi} p_{x*} r_j \sin \theta d\theta dz & \int\limits_0^L \int\limits_0^{2\pi} p_{y*} r_j \sin \theta d\theta dz \end{pmatrix} \tag{17}$$

$$\Omega^2 M_{CR} = \frac{B_{xx} K_{zz} + B_{zz} K_{xx} - B_{zx} K_{xz} - B_{xz} K_{zx}}{B_{xx} + B_{zz}} \tag{18}$$

$$\Omega^2 = \frac{\left(K_{xx} - \Omega^2 M_{CR}\right)\left(K_{zz} - \Omega^2 M_{CR}\right) - K_{xz}K_{zx}}{B_{xx}B_{zz} - B_{xz} - B_{zx}} \tag{19}$$

*2.4. Method for Grey-Fuzzy Hybrid Optimization Model of Rough Elliptic Bore Misaligned Journal Bearing*

The present work utilized integrated algorithms of grey relational analysis and fuzzy logic to optimize the various responses. The process outline of this hybrid optimization is represented through a flowchart and is shown in Figure 2.

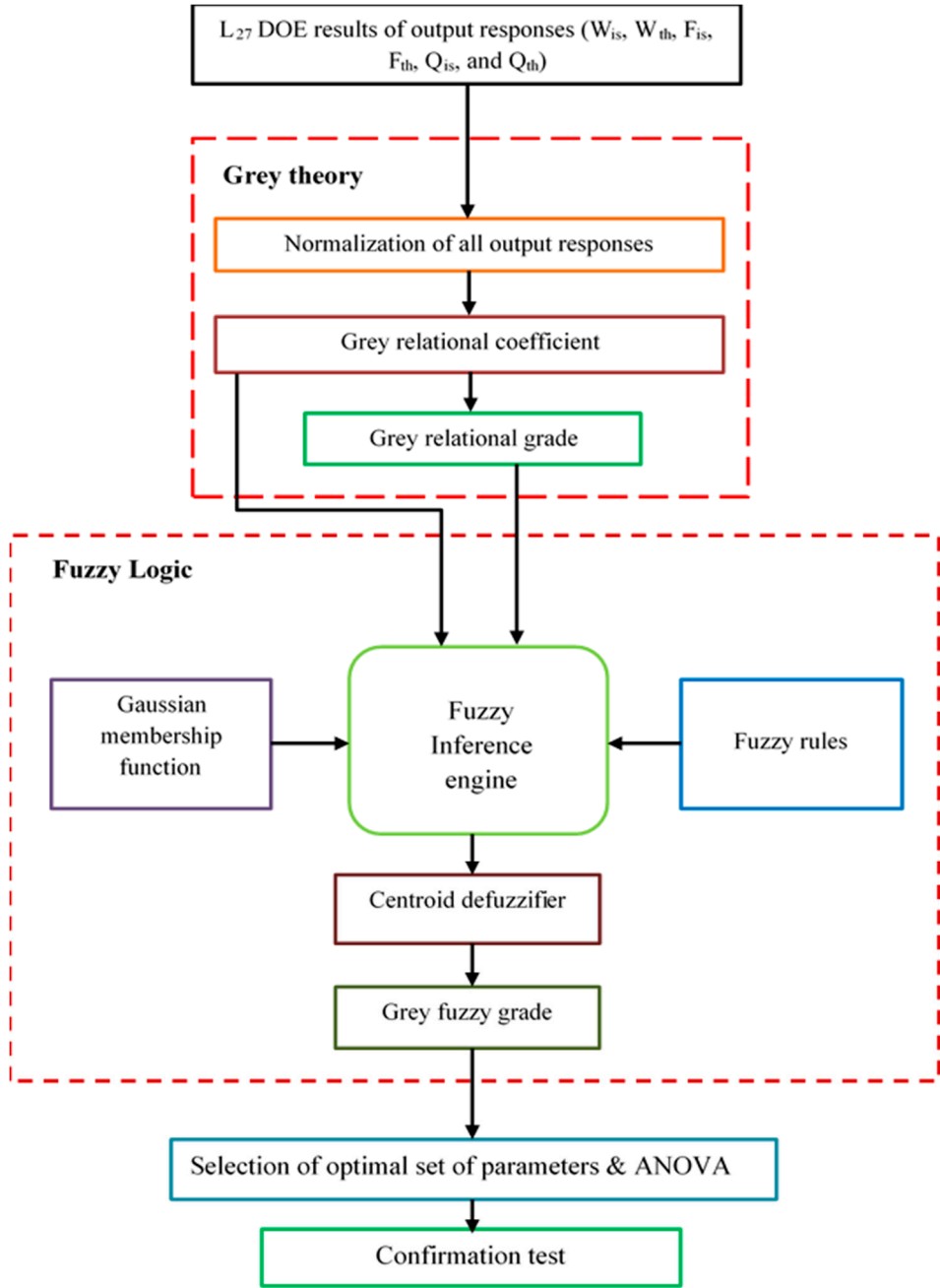

**Figure 2.** Flowchart of grey-fuzzy hybrid optimization.

In the grey theory, the term grey denotes the primitive data with deprived, incomplete, and unclear information. Moreover, it provides the incomplete connection of information among these data. Grey relational analysis (GRA) utilizes the quantitative exploration to define the degree of relationship among experimentally obtained results and a target value

in the grey system. The extent of relationship among these sequences can be expressed as the grey relational coefficient. In the grey relational analysis, the raw data of $W_{is}$, $W_{th}$, $F_{is}$, $F_{th}$, $Q_{is}$, and $Q_{th}$ are initially normalized in the range between 0 and 1 (four avoiding the unit of responses) and this process is known as the grey relational generation. For normalization, the responses, such as $W_{is}$, $W_{th}$, $Q_{is}$, and $Q_{th}$ have followed the larger-the-better characteristics (Equation (20)), while the other responses ($F_{is}$ and $F_{th}$) have followed the smaller-the-better characteristics (Equation (21)).

$$X_i(x) = \frac{Z_i(x) - \min Z_i(x)}{\max Z_i(x) - \min Z_i(x)} \tag{20}$$

$$X_i(x) = \frac{\max Z_i(x) - Z_i(x)}{\max Z_i(x) - \min Z_i(x)} \tag{21}$$

where '$X_i(x)$' is the normalized data of $i$th experiment for $x$th response, $Z$ is the response data, '$i$' symbolizes the number of experiments, i.e., $i = 1, 2, 3, \ldots, 27$, '$x$' is the number of response, i.e., $x = 1, 2, 3, \ldots, 6$, and 'min $Z_i(x)$' and 'max $Z_i(x)$' are the least and largest value of '$Z_i(x)$' for $x$th response, respectively.

Furthermore, considering normalized data, the grey relational coefficient (GRC) for each response is evaluated using Equation (22).

$$C_i(x) = \frac{K_{\min} + \lambda K_{\max}}{K_{0i}(x) + \lambda K_{\max}} \tag{22}$$

where $K_{0i}(x) = \|\varepsilon_0(x) - \varepsilon_i(x)\|$ is the absolute value of difference between the $K_0(x)$ and $K_i(x)$, $K_{\min}$ and $K_{\max}$ are the least and largest value of $K_{0i}(x)$ independently, and '$\lambda$' is a distinguishing term which lies between 0 and 1. In the current work, '$\lambda$' is considered as 0.5.

Furthermore, the grey relational grade (GRG) is a single response data that is estimated using Equation (23).

$$G_i = \frac{1}{m} \sum_{x=1}^{m} \lambda_i(x) \tag{23}$$

where '$G_i$' is the grey relational grade for $i$th experiment and $m$ denotes the number of response characteristics.

The fuzzy logical system is an artificial intelligence technique that can deal with uncertainty and anticipate future outcomes [28]. The fuzzy model comprises the fuzzifier, modem's membership relations, rule inference (as shown in Figure 3), inference engine, and de-fuzzifier. Initially, the fuzzifier applies the modem's membership relations to fuzzify the grey relational coefficients (GRC). The membership function (MF) defines the mapping of inputs and outputs. Several types of MF, such as Trapezoidal, Sigmoidal, Gaussian, and Triangular have been found in the literature. Among these types, Gaussian membership function is the most popular due to several advantageous features, such as local although not stringently compact, exhibiting a very smooth output, and predictions that are non-probabilistic [12]. Therefore, Gaussian membership function is utilized here to achieve the grey-fuzzy grade (GFG) value. Seven numbers of MF (VVL—very very low, VL—very low, L—low, M—medium, H—high, VH—very high, and VVH—very very high) are considered for each input and output. Moreover, the membership graph for each input and output is displayed in Figures 4 and 5, respectively. Furthermore, 20 different fuzzy rules are utilized to achieve the predicted data of GFG. To de-fuzzify the output data, most of the prevailing centroid technique is utilized to achieve more crispy output results [29]. The fuzzy rules viewer is utilized to achieve the GFG data for all the experimental runs. All these activities for predicting GFG are accomplished using the '*fuzzy*' tool available in Matlab-R2013a software from MathWorks.

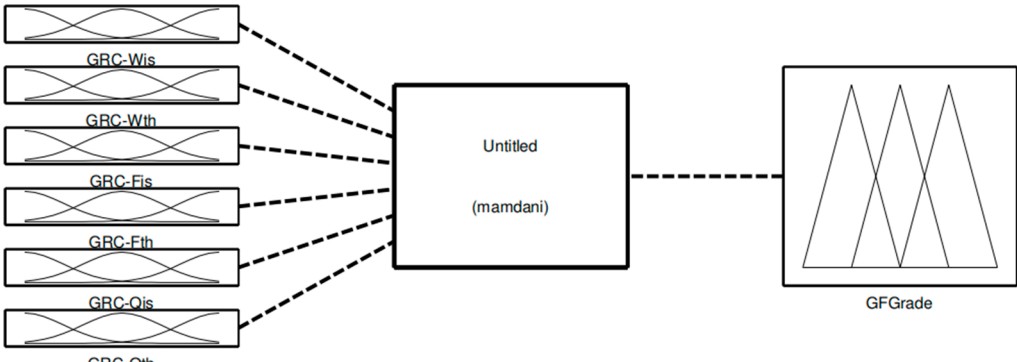

**Figure 3.** Fuzzy inference system.

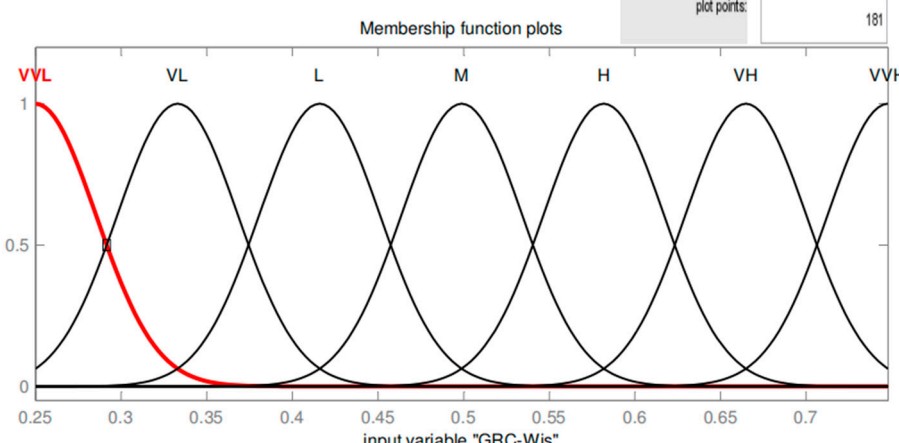

**Figure 4.** The membership graph of input.

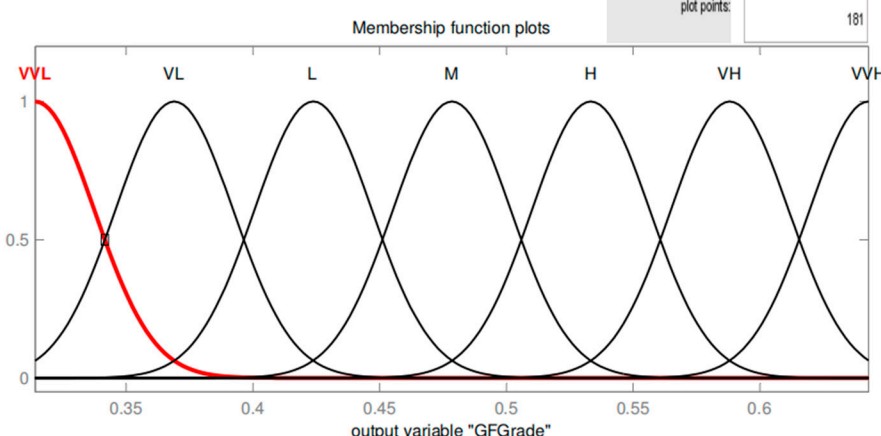

**Figure 5.** The membership graph of output.

## 3. Results and Discussion

### 3.1. Rough Elliptic Bore Misaligned Journal Bearing Static Performance Analysis

Due to the presence of geometrical abnormalities, the film profile varies accordingly. In the case of an elliptic bore, two conjunctions (converging diverging zones) exist with two minima, which are symmetrically located with respect to the mean position, as compared to the single point minima in the case of circular bore [16]. Furthermore, a circular bore with misalignment bears a single point of minima in arbitrary positions, while the induction of roughness creates micro-conjunctions of 0.2 μm in amplitude.

The film thickness and hydrodynamic pressure profile in three-dimensional form is shown in Figure 6a,b, respectively. In Equation (3), the hydrodynamic pressure is more

evenly distributed in the case of an elliptic bore bearing without misalignment compared with misalignment. In the former case, there is a 20% increase in pressure spike, whereas in the case of a circular misaligned case, there is a concentrated pressure which may lead to a metal-to-metal contactor with a higher degree of instability. The rough elliptic bore misaligned journal bearing configuration provides a more distributed pressure profile compared with the misaligned smooth elliptic bore, although the peak hydrodynamic pressure is reduced to 33%. However, the bearing stability is better in the former case.

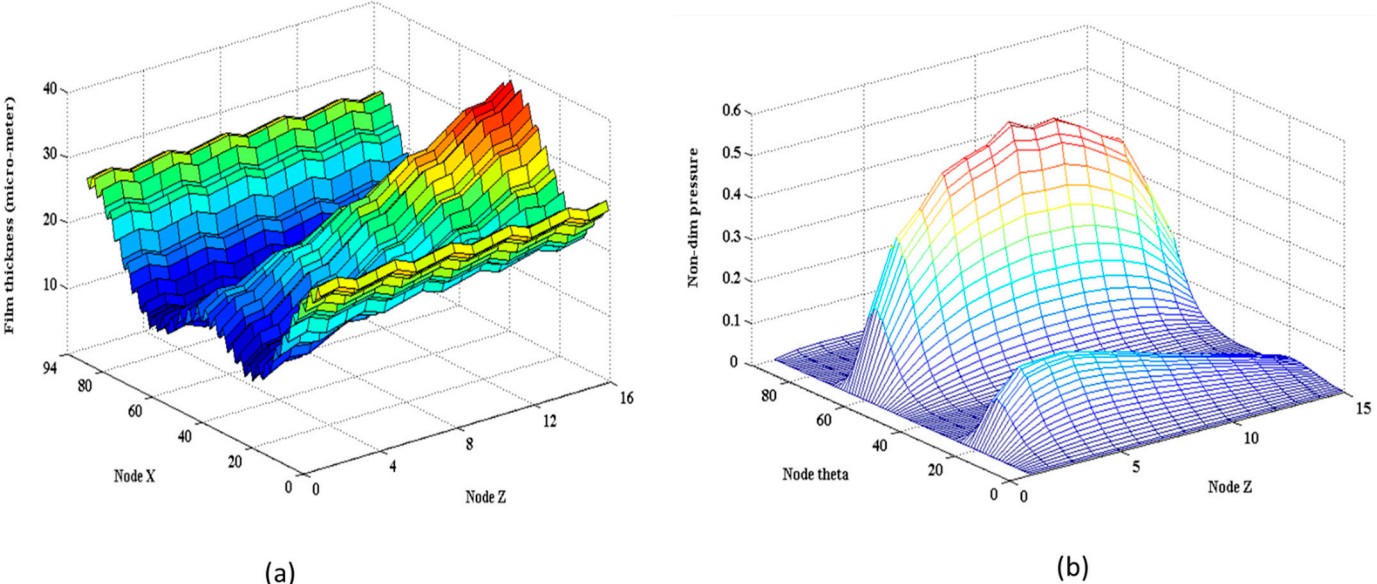

(a)                                                                    (b)

**Figure 6.** Misaligned rough elliptic bore journal bearing. (**a**) Three-dimensional film profile [18]; (**b**) three-dimensional pressure profile [18].

The rapid shear of the lubricant layer develops a temperature in the film. Therefore, the developed heat was transferred across the film to the rotating shaft and stationary bore through the conduction mode. The viscosity and density dependence of temperature is shown in Equation (3a,b). For G = 1.0, $\varepsilon$ = 0.6, and $D_m$ = 0, the isotherms are regular with an 8 °C difference in temperature. Figure 7a,b shows the temperature contour of bearing surface for the longitudinal and transverse pattern of roughness, respectively, with the highest temperature as 47 °C. At the angular location of minimum film thickness, with a degree of misalignment of $D_m$ = 0.8, the trend of the isotherm changed (moved to a different location) and the maximum temperature was reduced to 43 °C in the corresponding region. Reduction in temperature is due to turbulence in the conjunction as a result of rough-elliptic-misaligned geometry. The presence of transverse/longitudinal roughness leads to a variation in higher order isotherms and their contour profile on the bearing surface. Although no significant profile variation on lower order isotherms exists, the misaligned rough elliptic bore journal bearing has dynamic characteristics.

With the increasing eccentricity ratio, $K_{xx}$ decreases by 62% between 0.3 < and < 0.9. For a particular eccentricity in this range, the $K_{xx}$ is higher for a higher degree of misalignment, with is a 68% increase in $K_{xx}$ for $\Delta D_m$ = 0.9. The power index (PI) has little or no effect on $K_{xx}$. However, for a higher degree of misalignment, the $K_{xx}$ is higher for any PI. The highest value of $K_{xx}$ of 0.61 occurs at a PI of 0.6 and $D_m$ = 0.0. Similarly, the angular speed has little or no effect on $K_{xx}$. The maximum $K_{xx}$ is 0.61 at 3000 rpm for 0.9 degrees of misalignment ($D_m$). At particular G, $K_{xx}$ increases with the increase in $D_m$. For $\Delta D_m$ = 0.9, a 76.4% increase in $K_{xx}$ is observed. The increment follows a parabolic trend, with $K_{xx}$ having a lower value at the higher value of G.

The $K_{xz}$ and $K_{zx}$ have a reversing trend. The $K_{xz}$ increases with an increasing eccentricity ratio; ($dK_{xz}/d\varepsilon$) = 0.45 for $D_m$ = 0; ($dK_{xz}/d\varepsilon$) = 0.56 for $D_m$ = 0.5; and ($dK_{xz}/d\varepsilon$) = 0.63 for Dm = 0.9. Similar to $K_{xx}$, $K_{xz}$ has negligible change with an increase in PI and angular

speed (N) of the shaft. At particular PI, $(K_{xz})_{Dm=0} > (K_{xz})_{Dm=0.5} > (K_{xz})_{Dm=0.9}$. The $K_{xz}$ response to $D_m$ follows a parabolic trend with various ellipticities (G). For a particular non-circularity, $K_{xz}$ increases with the increasing $D_m$. At a certain degree of misalignment $(D_m)$, $K_{xz}$ is higher for higher non-circularity. Details of all dynamic stability parameters are shown in Tables 3–6. Variations of $K_{xx}$, $K_{xz}$, $K_{zx}$, $K_{zz}$, $B_{xx}$, $B_{xz}$, $B_{zx}$, $B_{zz}$, $M_{cr}$, and $\Omega$ for various combinations of $(\varepsilon, D_m)$, $(PI, D_m)$, $(N, D_m)$, and $(G, D_m)$ are stated in these tables.

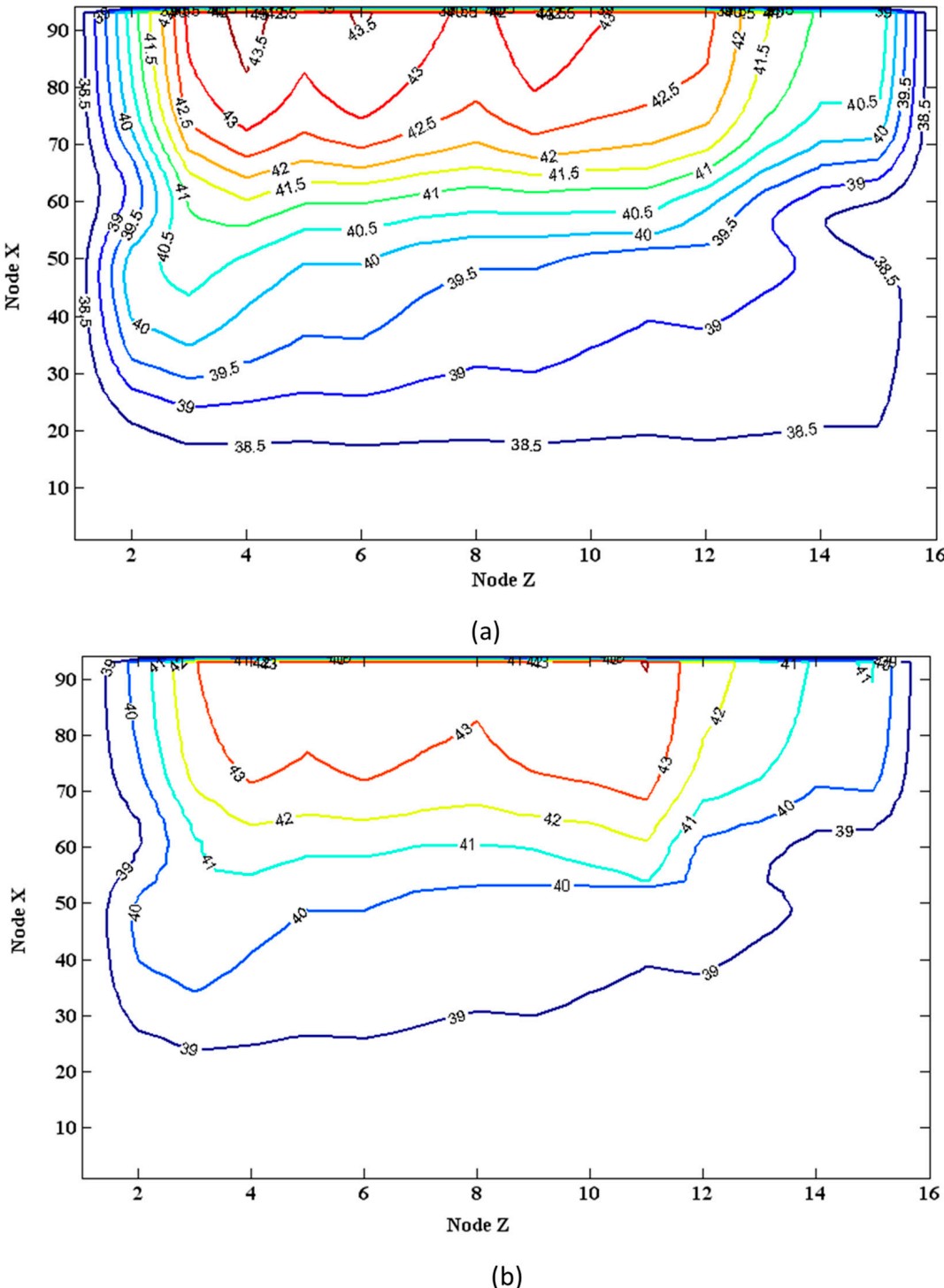

**Figure 7.** Misaligned rough elliptic bore journal bearing temperature counter. (**a**) Longitudinal pattern [18]; (**b**) transverse pattern [18].

The bearing stiffness coefficient along the z-z direction ($K_{zz}$) increases with an increase in misalignment. At $\varepsilon_{0.3}$ for $\Delta D_{m0.9}$, a 73% increase in $K_{zz}$ is observed. Similarly, at $\varepsilon_{0.6}$, it is increased by 90% due to the degree of misalignment of $\Delta D_{m0.9}$. It increases further at $\varepsilon_{0.9}$ for $\Delta D_{m0.9}$ and is found to be 100%. Due to the combined effect of PI and misalignment, $K_{zz}$ increases with the increasing value of $D_m$ at particular $PI_{0.2/0.34/0.6}$. For $PI_{0.2}$ and $\Delta D_{m0.9}$, the increase in $K_{zz}$ is observed to be 89.6%. Similarly, for $PI_{0.34}$ and $\Delta D_{m0.9}$, an 86.3% increase in $K_{zz}$ is observed. At the highest power index ($PI_{0.9}$) and $\Delta D_{m0.9}$, $K_{zz}$ increases by 86.1%. At the highest $D_{m0.9}$, for $\Delta P_{0.2-0.34}$, $K_{zz}$ decreases by 10.3%.

**Table 3.** Combined eccentricity and misalignment effect on stability parameters.

|  | $K_{xx}$ | $K_{xz}$ | $K_{zx}$ | $K_{zz}$ | $B_{xx}$ | $B_{xz}$ | $B_{zx}$ | $B_{zz}$ | $M_{cr}$ | $\Omega$ |
|---|---|---|---|---|---|---|---|---|---|---|
| $\varepsilon$ (0.3), $D_m$ (0) | 0.45 | −1.1 | 3.2 | 1.5 | 5.0 | −2.4 | −4.8 | 2.8 | 3.3 | 0.145 |
| $\varepsilon$ (0.3), $D_m$ (0.5) | 0.33 | −0.8 | 5.1 | 2.4 | 8.5 | −3.7 | −7.6 | 4.5 | 5.3 | 0.145 |
| $\varepsilon$ (0.3), $D_m$ (0.9) | 0.22 | −0.45 | 5.4 | 2.6 | 9.1 | −4.0 | −8.1 | 5.0 | 5.6 | 0.145 |
| $\varepsilon$ (0.6), $D_m$ (0) | 0.75 | −0.9 | 2.1 | 1.0 | 3.0 | −1.7 | −3.3 | 1.9 | 2.3 | 0.145 |
| $\varepsilon$ (0.6), $D_m$ (0.5) | 0.55 | −1.25 | 3.6 | 1.7 | 5.6 | −2.6 | −5.2 | 3.2 | 3.7 | 0.145 |
| $\varepsilon$ (0.6), $D_m$ (0.9) | 0.21 | −0.78 | 3.8 | 1.9 | 6.0 | −2.9 | −5.7 | 3.4 | 3.9 | 0.145 |
| $\varepsilon$ (0.9), $D_m$ (0) | 0.8 | −1.95 | 1.2 | 0.6 | 1.3 | −1.0 | −1.9 | 1.0 | 1.4 | 0.145 |
| $\varepsilon$ (0.9), $D_m$ (0.5) | 0.45 | −1.35 | 1.9 | 1.0 | 2.8 | −1.6 | −3.2 | 1.8 | 2.2 | 0.145 |
| $\varepsilon$ (0.9), $D_m$ (0.9) | 0.33 | −0.85 | 2.1 | 1.2 | 2.97 | −1.8 | −3.5 | 1.9 | 2.35 | 0.145 |

**Table 4.** Power index and misalignment effect on stability parameters.

|  | $K_{xx}$ | $K_{xz}$ | $K_{zx}$ | $K_{zz}$ | $B_{xx}$ | $B_{xz}$ | $B_{zx}$ | $B_{zz}$ | $M_{cr}$ | $\Omega$ |
|---|---|---|---|---|---|---|---|---|---|---|
| PI (0.2), $D_m$ (0) | 0.36 | −0.89 | 0.25 | 1.2 | 4.65 | −1.8 | −3.6 | 4.1 | 44 | 0.148 |
| PI (0.2), $D_m$ (0.5) | 0.35 | −1.25 | 0.41 | 2.0 | 7.54 | −2.7 | −5.8 | 3.8 | 41 | 0.148 |
| PI (0.2), $D_m$ (0.9) | 0.34 | −1.48 | 0.44 | 2.25 | 7.98 | −2.48 | −6.5 | 2.5 | 27.0 | 0.148 |
| PI (0.34), $D_m$ (0) | 0.56 | −0.9 | 0.23 | 1.1 | 4.8 | −1.7 | −3.8 | 4.0 | 45 | 0.148 |
| PI (0.34), $D_m$ (0.5) | 0.55 | −1.32 | 0.40 | 1.98 | 7.6 | −2.74 | −6.1 | 3.7 | 42 | 0.148 |
| PI (0.34), $D_m$ (0.9) | 0.52 | −1.4 | 0.45 | 2.05 | 8.1 | −2.5 | −6.6 | 2.4 | 26.3 | 0.148 |
| PI (0.6), $D_m$ (0) | 0.61 | −0.92 | 0.27 | 1.15 | 4.91 | −1.8 | −3.9 | 3.9 | 47 | 0.148 |
| PI (0.6), $D_m$ (0.5) | 0.60 | −1.50 | 0.43 | 1.96 | 7.63 | −2.8 | −6.2 | 3.6 | 42.5 | 0.148 |
| PI (0.6), $D_m$ (0.9) | 0.58 | −1.52 | 0.47 | 2.14 | 8.25 | −2.6 | −6.7 | 2.3 | 27.3 | 0.148 |

**Table 5.** Combined angular speed and misalignment effect on stability parameters.

|  | $K_{xx}$ | $K_{xz}$ | $K_{zx}$ | $K_{zz}$ | $B_{xx}$ | $B_{xz}$ | $B_{zx}$ | $B_{zz}$ | $M_{cr}$ | $\Omega$ |
|---|---|---|---|---|---|---|---|---|---|---|
| N (3000), $D_m$ (0) | 0.33 | −8.4 | 0.22 | 1.20 | 1.4 | −0.7 | −1.4 | 0.8 | 6 | 4.4 |
| N (3000), $D_m$ (0.5) | 0.30 | −13.65 | 0.35 | 1.85 | 2.3 | −0.95 | −1.9 | 1.3 | 5.2 | 4.4 |
| N (3000), $D_m$ (0.9) | 0.29 | −14.68 | 0.41 | 2.0 | 2.6 | −1.1 | −2.2 | 1.4 | 5.0 | 4.4 |
| N (6000), $D_m$ (0) | 0.53 | −8.7 | 0.25 | 1.25 | 3.1 | −1.2 | −2.7 | 1.6 | 11.0 | 2.2 |
| N (6000), $D_m$ (0.5) | 0.51 | −13.7 | 0.4 | 1.95 | 4.8 | −1.95 | −4.0 | 2.5 | 18.0 | 2.2 |
| N (6000), $D_m$ (0.9) | 0.49 | −14.7 | 0.43 | 2.12 | 5.2 | −2.2 | −4.7 | 2.7 | 20.0 | 2.2 |
| N (9000), $D_m$ (0) | 0.62 | −8.8 | 0.2 | 1.32 | 4.5 | −1.7 | −4.0 | 2.4 | 25.0 | 1.5 |
| N (9000), $D_m$ (0.5) | 0.59 | −13.72 | 0.34 | 1.98 | 7.3 | −3.0 | −6.1 | 3.7 | 41.0 | 1.5 |
| N (9000), $D_m$ (0.9) | 0.57 | −14.73 | 0.44 | 2.31 | 7.8 | −3.2 | −6.3 | 4.1 | 45.0 | 1.5 |

**Table 6.** Combined non-circularity and misalignment effect on stability parameters.

|                         | $K_{xx}$ | $K_{xz}$ | $K_{zx}$ | $K_{zz}$ | $B_{xx}$ | $B_{xz}$ | $B_{zx}$ | $B_{zz}$ | $M_{cr}$ | $\Omega$ |
|-------------------------|----------|----------|----------|----------|----------|----------|----------|----------|----------|----------|
| G (1.0), $D_m$ (0)      | 0.18     | −0.5     | 0.97     | 3.3      | 2.0      | −0.8     | −1.8     | 5.9      | 45       | 0.215    |
| G (1.0), $D_m$ (0.5)    | 0.26     | −0.07    | 1.04     | 5.0      | 2.3      | −0.9     | −2.0     | 6.4      | 49       | 0.215    |
| G (1.0), $D_m$ (0.9)    | 0.28     | −0.78    | 1.08     | 5.3      | 2.4      | −1.2     | −2.2     | 6.6      | 50       | 0.215    |
| G (2.0), $D_m$ (0)      | 0.36     | −0.95    | 0.36     | 1.0      | 4.5      | −1.8     | −3.7     | 2.3      | 17       | 0.215    |
| G (2.0), $D_m$ (0.5)    | 0.58     | −1.5     | 0.41     | 1.7      | 5.0      | −1.9     | −4.1     | 2.5      | 19       | 0.215    |
| G (2.0), $D_m$ (0.9)    | 0.6      | −1.57    | 0.44     | 2.1      | 5.2      | −2.0     | −4.3     | 2.7      | 20       | 0.215    |
| G (3.0), $D_m$ (0)      | 0.98     | −2.5     | 0.12     | 0.6      | 12.0     | −4.8     | −9.7     | 1.1      | 8        | 0.215    |
| G (3.0), $D_m$ (0.5)    | 1.46     | −3.6     | 0.19     | 1.1      | 12.7     | −4.9     | −10.3    | 1.3      | 9        | 0.215    |
| G (3.0), $D_m$ (0.9)    | 1.5      | −3.8     | 0.21     | 1.34     | 13.2     | −5.1     | −10.6    | 1.4      | 10       | 0.215    |

The combined effect of angular speed and misalignment shows that for $\Delta N_{3000-9000}$, $K_{zz}$ increases by 10%. For aligned bearing ($D_{m0.0}$) at 3000 rpm due to the first 50% increase in misalignment ($\Delta D_{m0.0-0.5}$), $K_{zz}$ increases by 66.7%. At this angular speed, which is further due to $\Delta D_{m0.5-0.9}$, the $K_{zz}$ increases by 8.1%. Similarly, at 6000 rpm, $K_{zz}$ increases by 56% for a misalignment difference of $\Delta D_{m0.0-0.5}$. Once again, for $\Delta D_{m0.5-0.9}$, an 8.7% increase in $K_{zz}$ is observed for the same angular speed. At 9000 rpm, the lower-level change in misalignment $\Delta D_{m0.0-0.5}$, induces 50% more $K_{zz}$, whereas in the upper-level change in misalignment $\Delta D_{m0.5-0.9}$, it increases by 16.7%. At a particular degree of misalignment, $D_{m0/0.5/0.9}$, $K_{zz}$ increases with the increasing angular speed. At $D_{m0}$, for $\Delta N_{3000-6000}$ and $\Delta N_{6000-9000}$, an increase in $K_{zz}$ is observed to be 4.17% and 5.6%, respectively. For an aligned bearing ($D_{m0.0}$), $K_{zz}$ decreases with an increase in non-circularity range, $\Delta G_{1.0-2.0}$, and $\Delta G_{2.0-3.0}$, at a rate of 60% and 40%, respectively.

The misaligned cases exhibit a similar trend to the aligned cases. For $G_{1.0}$, at $\Delta D_{m0.0-0.5}$, $K_{zz}$ increases by 34%, whereas it increases by 6% for $\Delta D_{m0.5.0-0.9}$ misalignment increment. For $G_{2.0}$, at $\Delta D_{m0.0-0.5}$ and $\Delta D_{m0.5-0.9}$, $K_{zz}$ is observed to increase by 70% and 23.5%, respectively. For higher non-circularity $G_{3.0}$ and lower-level increment of $\Delta D_{m0.0-0.5}$, $K_{zz}$ is observed to increase by 83.3%, whereas for higher-level increment of $\Delta D_{m0.5-0.9}$, it increases by 21.8%. The combination of lower non-circularity and misalignment elevates $K_{zz}$, while the higher non-circularity and misalignment has a diminishing effect on $K_{zz}$. There are four damping coefficients ($B_{xx}$, $B_{xz}$, $B_{zx}$, $B_{zz}$) associated with the bearing stability analysis, among which $B_{xx}$ and $B_{zz}$ are considered positive, while $B_{xz}$ and $B_{zx}$ are considered negative. The highest value of $B_{xx}$ is 13.2 at $G_{3.0}$ and $D_{m0.9}$, whereas the lowest is 1.4 at $N_{3000}$ and $D_{m0.0}$. At a particular eccentricity ratio ($\varepsilon_{0.3/0.6/0.9}$), $B_{xx}$ increases with the increasing misalignment. At lower eccentricity, $\varepsilon_{0.3}$, for $\Delta D_{m0.0-0.5}$, $B_{xx}$ is observed to increase by 70%, while for the same eccentricity and upper misalignment range, $\Delta D_{m0.5-0.9}$, a 7% increase in $B_{xx}$ is observed. At medium eccentricity, $\varepsilon_{0.6}$, a 7.1% increase in $B_{xx}$ is observed for $\Delta D_{m0.0-0.5}$ and $\Delta D_{m0.5-0.9}$, respectively. At higher eccentricity, $\varepsilon_{0.9}$, $B_{xx}$ growth is lowest. The rate of growth is 115% and 6%, respectively for the misalignment range of $\Delta D_{m0.0-0.5}$ and $\Delta D_{m0.5-0.9}$.

For aligned shafts, $D_{m0.0}$, $B_{xx}$ decreases slightly for a range of $PI_{0.2-0.34-0.6}$. At a particular $PI_{0.2/0.34/0.6}$, for a lower range of misalignment $\Delta D_{m0.0-0.5}$, $B_{xx}$ is observed to increase by 62.1%, 58.4%, and 55.4% for the respective power index. Meanwhile, for higher misalignment range $\Delta D_{m0.5-0.9}$, $B_{xx}$ is observed to increase by 5.8%, 6.5%, and 8.1%. $B_{xx}$ increases with the combined effect of angular speed and misalignment. The lowest is 1.4 at $N_{3000}$ and $D_{m0.0}$, whereas the highest is 7.8 at $N_{9000}$ and $D_{m0.9}$. $B_{xx}$ increases with the combined effect of non-circularity ($G_{1.0-2.0-3.0}$) and misalignment ($D_{m0.0-0.5-0.9}$). The lowest is 2.0 at $G_{1.0}$ and $D_{m0.0}$, whereas the highest is 13.2 at $G_{3.0}$ and $D_{m0.9}$.

The damping coefficients, $B_{xz}$ and $B_{zx}$, have a negative trend and increase with the combined effect of angular speed and misalignment, the lowest being −0.7 and −1.4,

respectively, at 3000 rpm at $D_{m0.0}$ and the highest being $-3.2$ and $-6.3$, respectively. The next important damping coefficient is $B_{zz}$, acting in the z-z direction. $B_{zz}$ increases with the increasing $D_{m0.0-0.5-0.9}$ for lower eccentricities. At lower eccentricity, $\varepsilon_{0.3}$, for $\Delta D_{m0.0-0.5}$, $B_{zz}$ is observed to increase by 60.7%, while for the same eccentricity and upper misalignment range, $\Delta D_{m0.5-0.9}$, an 11.1% increase in $B_{zz}$ is observed. For aligned shafts, $B_{zz}$ decreases with an increase in $\varepsilon_{0.3-0.6-0.9}$. Due to the combined effect of $\varepsilon$ and $D_m$ changes, the highest value of $B_{zz}$ is 5.0 at $\varepsilon_{0.3}$ and $D_{m0.9}$. At a particular power index ($PI_{0.2-0.34-0.6}$), $B_{zz}$ decreases with an increasing misalignment degree. The highest value of $B_{zz}$ is 4.1 at $PI_{0.2}$ and $D_{m0.0}$. The lowest value of $B_{zz}$ is 2.3 at $PI_{0.6}$ and $D_{m0.9}$. Due to the combined effect of angular speed and misalignment, $B_{zz}$ increases $N_{3000-6000-9000}$ and $D_{m0.0-0.5-0.9}$. The highest $B_{zz}$ occurs in this condition at $N_{9000}$ and $D_{m0.9}$, whereas the lowest is 0.8 at $N_{3000}$ and $D_{m0.0}$. For both aligned and misaligned cases, $B_{zz}$ increases with an increase in $D_{m0.0-0.5-0.9}$ and particular non-circularity, $G_{1.0/2.0/3.0}$. For the combined effect of non-circularity and misalignment, the highest value of $B_{zz}$ is observed at $G_{1.0}$ and $D_{m0.0}$, while the lowest is found to be 1.1 at $G_{3.0}$ and $D_{m0.0}$.

In bearing stability consideration, critical mass ($M_{cr}$) is one important parameter. At a particular degree of misalignment, $D_{m0.0-0.5-0.9}$, $M_{cr}$ decreases with an increase in eccentricities, $\varepsilon_{0.3-0.6-0.9}$. The highest $M_{cr}$ is observed to be 5.6 at $\varepsilon_{0.3}$ and $D_{m0.9}$, whereas the lowest is 1.4 at $\varepsilon_{0.9}$ and $D_{m0.0}$. Due to the combined effect of power index and misalignment, $M_{cr}$ increases with an increase in $PI_{0.2-0.34-0.6}$ for a particular degree of misalignment, $D_{m0.0/0.5/0.9}$. In this case, the highest value of $M_{cr}$ is 47 at $PI_{0.6}$ and $D_{m0.0}$, whereas the lowest is found to be 26.3 at $PI_{0.34}$ and $D_{m0.9}$. The combined effect of angular speed and misalignment leads to the highest value that elevates the $M_{cr}$. The highest value of $M_{cr}$ at this condition is 45 for $N_{9000}$ and $D_{m0.9}$, whereas the lowest is 6 for $N_{3000}$ and $D_{m0.0}$. The combined effect of non-circularity and misalignment has a reducing effect on $M_{cr}$. In this specific condition, the highest value of $M_{cr}$ occurs at $G_{1.0}$ and $D_{m0.0}$, whereas the lowest value of $M_{cr}$ is 8 at $G_{3.0}$ and $D_{m0.0}$.

Whirl ratio ($\Omega$) is the last stability parameter, which fluctuates more due to the combined effect of angular speed and misalignment. The highest value of $\Omega$ was observed to be 4.4 at $N_{3000}$ for all $D_{m0.0-0.5-0.9}$, whereas the lowest is 1.5 at $N_{9000}$ for all $D_{m0.0-0.5-0.9}$. For other combinations of parameters, the whirl ratio is estimated as 0.145, 0.148, and 0.215, respectively.

*3.2. Grey-Fuzzy Hybrid Result of Rough Elliptic Bore Misaligned Journal Bearing*

The orthogonal array for design of experiment (DOE) $L_{27}$ deals with four input parameters and six output parameters [30], as shown in Table 7. The eccentricity ($\varepsilon$) of 0.3, 0.5, and 0.8; the L/D ratio ($\beta$) of 0.5, 1.0, and 2.0; the non-circularity (G) of 0.5, 1.0, and 2.0; and the roughness coefficient(y) of 0.1, 0.2, and 0.3 are considered as input levels for the development of the DOE for this particular analysis. Tables 8 and 9 shows the estimated GRC, GRG, and GFG values, and fuzzy rules respectively.

Figure 8 shows the plot for GFG. Furthermore, the mean GFG is estimated with respect to the input factors and their levels as displayed in Table 10. The delta value for each factor is estimated to know the impact of input terms on GFG. Based on delta values, the rank has been determined. The largest magnitude of delta confirms the highest rank and similarly, the ranks are estimated for each input factor. From these statistics, the factor ($\varepsilon$) exhibited the largest impact on output (GFG), while 'y' exhibited the least impact. A larger mean value of GFG of individual input indicates that the optimal level of corresponding factors is $\varepsilon_1$ (0.3)-$\beta_1$ (0.5)-$G_3$ (2)-$y_1$ (0.1). Table 11 show the percentage contribution of input terms, while Table 12 shows optimal and predicted results.

**Table 7.** Orthogonal array $L_{27}$ of the simulation [30] runs with results (at $D_m$ = 0.9).

| Run | Input Variables | | | | Performance Characteristics | | | | | |
|-----|-----------------|---|---|---|-----------------------------|---|---|---|---|---|
| | Factor-1 ($\varepsilon$) | Factor-2 ($\beta$) | Factor-3 (G) | Factor-4 (y) | $W_{is}$ | $W_{th}$ | $F_{is}$ | $F_{th}$ | $Q_{is}$ | $Q_{th}$ |
| 1 | 0.3 | 0.5 | 0.5 | 0.1 | 3.9176 | 1.302 | 52.95 | 22.51 | 4.4394 | 5.2438 |
| 2 | 0.3 | 0.5 | 0.5 | 0.2 | 3.882 | 1.302 | 33.85 | 14.83 | 4.4388 | 5.2429 |
| 3 | 0.3 | 0.5 | 0.5 | 0.3 | 3.8245 | 1.3026 | 33.73 | 14.83 | 4.4382 | 5.2415 |
| 4 | 0.3 | 1.0 | 1.0 | 0.1 | 5.806 | 3.432 | 88.09 | 46.01 | 4.8875 | 5.8628 |
| 5 | 0.3 | 1.0 | 1.0 | 0.2 | 5.7681 | 3.432 | 30.83 | 13.12 | 4.8875 | 5.8605 |
| 6 | 0.3 | 1.0 | 1.0 | 0.3 | 5.7063 | 3.432 | 30.12 | 13.12 | 4.8875 | 5.8567 |
| 7 | 0.3 | 2.0 | 2.0 | 0.1 | 5.4342 | 3.214 | 94.27 | 58.46 | 5.2134 | 6.342 |
| 8 | 0.3 | 2.0 | 2.0 | 0.2 | 5.41 | 3.321 | 38.32 | 15.43 | 5.1121 | 6.123 |
| 9 | 0.3 | 2.0 | 2.0 | 0.3 | 5.13 | 3.133 | 36.31 | 14.35 | 5.0032 | 6.0021 |
| 10 | 0.5 | 0.5 | 1.0 | 0.1 | 5.6321 | 1.7324 | 102.67 | 42.15 | 5.5016 | 7.0872 |
| 11 | 0.5 | 0.5 | 1.0 | 0.2 | 5.5731 | 1.7324 | 33.53 | 12.98 | 5.5016 | 7.0863 |
| 12 | 0.5 | 0.5 | 1.0 | 0.3 | 5.4786 | 1.7324 | 32.71 | 12.98 | 5.5016 | 7.0847 |
| 13 | 0.5 | 1.0 | 2.0 | 0.1 | 5.9403 | 2.1872 | 95.36 | 36.65 | 5.324 | 6.832 |
| 14 | 0.5 | 1.0 | 2.0 | 0.2 | 5.9018 | 2.3413 | 40.32 | 14.32 | 5.123 | 6.342 |
| 15 | 0.5 | 1.0 | 2.0 | 0.3 | 5.81 | 2.43 | 38.43 | 13.78 | 5.006 | 6.213 |
| 16 | 0.5 | 2.0 | 0.5 | 0.1 | 10.5943 | 5.397 | 63.31 | 54.38 | 5.5016 | 5.7148 |
| 17 | 0.5 | 2.0 | 0.5 | 0.2 | 10.465 | 5.3972 | 38.23 | 31.18 | 5.5016 | 5.7116 |
| 18 | 0.5 | 2.0 | 0.5 | 0.3 | 10.2577 | 5.3972 | 38.09 | 31.18 | 5.5016 | 5.7063 |
| 19 | 0.8 | 0.5 | 2.0 | 0.1 | 8.8186 | 4.345 | 68.231 | 40.824 | 5.756 | 5.821 |
| 20 | 0.8 | 0.5 | 2.0 | 0.2 | 8.6272 | 4.234 | 40.13 | 40.021 | 5.754 | 5.281 |
| 21 | 0.8 | 0.5 | 2.0 | 0.3 | 8.2216 | 4.132 | 40.012 | 38.293 | 5.345 | 5.221 |
| 22 | 0.8 | 1.0 | 0.5 | 0.1 | 24.5895 | 6.19 | 74.78 | 31.49 | 5.9745 | 6.0376 |
| 23 | 0.8 | 1.0 | 0.5 | 0.2 | 23.8265 | 6.19 | 42.39 | 17.11 | 5.9745 | 6.0356 |
| 24 | 0.8 | 1.0 | 0.5 | 0.3 | 22.7026 | 6.19 | 40.52 | 17.11 | 5.9745 | 6.0324 |
| 25 | 0.8 | 2.0 | 1.0 | 0.1 | 17.9991 | 6.3785 | 17.9991 | 6.3785 | 6.4226 | 6.2705 |
| 26 | 0.8 | 2.0 | 1.0 | 0.2 | 17.6003 | 6.3785 | 17.6003 | 6.3785 | 6.4226 | 6.2675 |
| 27 | 0.8 | 2.0 | 1.0 | 0.3 | 16.7014 | 6.3785 | 16.7014 | 6.3785 | 6.4226 | 6.2625 |

**Table 8.** Estimated GRC, GRG, and GFG values.

| Sequential Order | Grey Relation Coefficient (GRC) | | | | | | GRG | GFG |
|------------------|---------------------------------|---|---|---|---|---|-----|-----|
| | $W_{is}$ | $W_{th}$ | $F_{is}$ | $F_{th}$ | $Q_{is}$ | $Q_{th}$ | | |
| 1 | 0.748 | 0.750 | 0.461 | 0.405 | 0.750 | 0.744 | 0.643 | 0.622 |
| 2 | 0.749 | 0.750 | 0.350 | 0.331 | 0.750 | 0.744 | 0.612 | 0.625 |
| 3 | 0.750 | 0.750 | 0.349 | 0.331 | 0.750 | 0.745 | 0.612 | 0.625 |
| 4 | 0.702 | 0.540 | 0.665 | 0.630 | 0.637 | 0.578 | 0.626 | 0.622 |
| 5 | 0.703 | 0.540 | 0.332 | 0.315 | 0.637 | 0.579 | 0.518 | 0.533 |
| 6 | 0.705 | 0.540 | 0.328 | 0.315 | 0.637 | 0.580 | 0.517 | 0.533 |
| 7 | 0.711 | 0.562 | 0.701 | 0.750 | 0.555 | 0.450 | 0.621 | 0.619 |
| 8 | 0.712 | 0.551 | 0.376 | 0.337 | 0.580 | 0.508 | 0.511 | 0.533 |
| 9 | 0.719 | 0.570 | 0.364 | 0.327 | 0.608 | 0.541 | 0.521 | 0.533 |
| 10 | 0.706 | 0.708 | 0.750 | 0.593 | 0.482 | 0.250 | 0.582 | 0.586 |
| 11 | 0.708 | 0.708 | 0.348 | 0.313 | 0.482 | 0.250 | 0.468 | 0.479 |
| 12 | 0.710 | 0.708 | 0.343 | 0.313 | 0.482 | 0.251 | 0.468 | 0.479 |
| 13 | 0.699 | 0.663 | 0.707 | 0.541 | 0.527 | 0.318 | 0.576 | 0.519 |
| 14 | 0.700 | 0.648 | 0.387 | 0.326 | 0.577 | 0.450 | 0.515 | 0.533 |
| 15 | 0.702 | 0.639 | 0.376 | 0.321 | 0.607 | 0.484 | 0.522 | 0.533 |
| 16 | 0.587 | 0.347 | 0.521 | 0.711 | 0.482 | 0.618 | 0.544 | 0.533 |

**Table 8.** *Cont.*

| Sequential Order | Grey Relation Coefficient (GRC) | | | | | | GRG | GFG |
|---|---|---|---|---|---|---|---|---|
| | $W_{is}$ | $W_{th}$ | $F_{is}$ | $F_{th}$ | $Q_{is}$ | $Q_{th}$ | | |
| 17 | 0.590 | 0.347 | 0.375 | 0.488 | 0.482 | 0.619 | 0.483 | 0.482 |
| 18 | 0.595 | 0.347 | 0.374 | 0.488 | 0.482 | 0.620 | 0.484 | 0.482 |
| 19 | 0.630 | 0.450 | 0.550 | 0.581 | 0.418 | 0.589 | 0.536 | 0.486 |
| 20 | 0.634 | 0.461 | 0.386 | 0.573 | 0.418 | 0.734 | 0.535 | 0.533 |
| 21 | 0.644 | 0.471 | 0.386 | 0.556 | 0.522 | 0.750 | 0.555 | 0.532 |
| 22 | 0.250 | 0.269 | 0.588 | 0.491 | 0.363 | 0.531 | 0.415 | 0.427 |
| 23 | 0.268 | 0.269 | 0.399 | 0.353 | 0.363 | 0.532 | 0.364 | 0.370 |
| 24 | 0.295 | 0.269 | 0.389 | 0.353 | 0.363 | 0.533 | 0.367 | 0.370 |
| 25 | 0.409 | 0.250 | 0.258 | 0.250 | 0.250 | 0.469 | 0.314 | 0.382 |
| 26 | 0.418 | 0.250 | 0.255 | 0.250 | 0.250 | 0.470 | 0.316 | 0.382 |
| 27 | 0.440 | 0.250 | 0.250 | 0.250 | 0.250 | 0.471 | 0.318 | 0.372 |

**Table 9.** Fuzzy rules.

| Rules Nos. | Description of Rules |
|---|---|
| 1 | If (GRC-$W_{is}$ is VVH) and (GRC-$W_{th}$ is VVH) and (GRC-$F_{is}$ is M) and (GRC-$F_{th}$ is L) and (GRC-$Q_{is}$ is VVH) and (GRC-$Q_{th}$ is VVH) then (GFGrade is VVH) |
| 2 | If (GRC-$W_{is}$ is VVH) and (GRC-$W_{th}$ is VVH) and (GRC-$F_{is}$ is VL) and (GRC-$F_{th}$ is VL) and (GRC-$Q_{is}$ is VVH) and (GRC-$Q_{th}$ is VVH) then (GFGrade is VVH) |
| 3 | If (GRC-$W_{is}$ is VH) and (GRC-$W_{th}$ is M) and (GRC-$F_{is}$ is VH) and (GRC-$F_{th}$ is VH) and (GRC-$Q_{is}$ is VH) and (GRC-$Q_{th}$ is H) then (GFGrade is VVH) |
| 4 | If (GRC-$W_{is}$ is VH) and (GRC-$W_{th}$ is M) and (GRC-$F_{is}$ is VL) and (GRC-$F_{th}$ is VL) and (GRC-$Q_{is}$ is VH) and (GRC-$Q_{th}$ is H) then (GFGrade is H) |
| 5 | If (GRC-$W_{is}$ is VVH) and (GRC-$W_{th}$ is H) and (GRC-$F_{is}$ is VH) and (GRC-$F_{th}$ is VVH) and (GRC-$Q_{is}$ is M) and (GRC-$Q_{th}$ is L) then (GFGrade is VVH) |
| 6 | If (GRC-$W_{is}$ is VVH) and (GRC-$W_{th}$ is H) and (GRC-$F_{is}$ is L) and (GRC-$F_{th}$ is VL) and (GRC-$Q_{is}$ is H) and (GRC-$Q_{th}$ is M) then (GFGrade is H) |
| 7 | If (GRC-$W_{is}$ is VH) and (GRC-$W_{th}$ is VH) and (GRC-$F_{is}$ is VVH) and (GRC-$F_{th}$ is H) and (GRC-$Q_{is}$ is VVL) and (GRC-$Q_{th}$ is VH) then (GFGrade is H) |
| 8 | If (GRC-$W_{is}$ is VH) and (GRC-$W_{th}$ is VH) and (GRC-$F_{is}$ is VL) and (GRC-$F_{th}$ is VL) and (GRC-$Q_{is}$ is M) and (GRC-$Q_{th}$ is VVL) then (GFGrade is M) |
| 9 | If (GRC- $W_{is}$ is VH) and (GRC-$W_{th}$ is VH) and (GRC-$F_{is}$ is L) and (GRC-$F_{th}$ is M) and (GRC-$Q_{is}$ is M) and (GRC-$Q_{th}$ is VL) then (GFGrade is VH) |
| 10 | If (GRC-$W_{is}$ is VH) and (GRC-$W_{th}$ is VH) and (GRC-$F_{is}$ is L) and (GRC-$F_{th}$ is VL) and (GRC-$Q_{is}$ is H) and (GRC-$Q_{th}$ is L) then (GFGrade is H) |
| 11 | If (GRC-$W_{is}$ is VH) and (GRC-$W_{th}$ is VH) and (GRC-$F_{is}$ is M) and (GRC-$F_{th}$ is VL) and (GRC-$Q_{is}$ is VH) and (GRC-$Q_{th}$ is L) then (GFGrade is H) |
| 12 | If (GRC-$W_{is}$ is H) and (GRC-$W_{th}$ is VL) and (GRC-$F_{is}$ is L) and (GRC-$F_{th}$ is VVH) and (GRC-$Q_{is}$ is M) and (GRC-$Q_{th}$ is VH) then (GFGrade is H) |
| 13 | If (GRC-$W_{is}$ is H) and (GRC-$W_{th}$ is VL) and (GRC-$F_{is}$ is L) and (GRC-$F_{th}$ is M) and (GRC-$Q_{is}$ is M) and (GRC-$Q_{th}$ is VH) then (GFGrade is M) |

**Table 9.** *Cont.*

| Rules Nos. | Description of Rules |
|---|---|
| 14 | If (GRC-$W_{is}$ is H) and (GRC-$W_{th}$ is VL) and (GRC-$F_{is}$ is M) and (GRC-$F_{th}$ is M) and (GRC-$Q_{is}$ is M) and (GRC-$Q_{th}$ is VH) then (GFGrade is M) |
| 15 | If (GRC-$W_{is}$ is VH) and (GRC-$W_{th}$ is L) and (GRC-$F_{is}$ is L) and (GRC-$F_{th}$ is H) and (GRC-$Q_{is}$ is L) and (GRC-$Q_{th}$ is H) then (GFGrade is H) |
| 16 | If (GRC-$W_{is}$ is VH) and (GRC-$W_{th}$ is M) and (GRC-$F_{is}$ is L) and (GRC-$F_{th}$ is H) and (GRC-$Q_{is}$ is L) and (GRC-$Q_{th}$ is VVH) then (GFGrade is H) |
| 17 | If (GRC-$W_{is}$ is VH) and (GRC-$W_{th}$ is M) and (GRC-$F_{is}$ is H) and (GRC-$F_{th}$ is H) and (GRC-$Q_{is}$ is M) and (GRC-$Q_{th}$ is VVH) then (GFGrade is VH) |
| 18 | If (GRC-$W_{is}$ is VVL) and (GRC-$W_{th}$ is VVL) and (GRC-$F_{is}$ is L) and (GRC-$F_{th}$ is M) and (GRC-$Q_{is}$ is L) and (GRC-$Q_{th}$ is M) then (GFGrade is L) |
| 19 | If (GRC-$W_{is}$ is VVL) and (GRC-$W_{th}$ is VVL) and (GRC-$F_{is}$ is L) and (GRC-$F_{th}$ is VL) and (GRC-$Q_{is}$ is L) and (GRC-$Q_{th}$ is M) then (GFGrade is VL) |
| 20 | If (GRC-$W_{is}$ is L) and (GRC-$W_{th}$ is VVL) and (GRC-$F_{is}$ is L) and (GRC-$F_{th}$ is VVL) and (GRC-$Q_{is}$ is VVL) and (GRC-$Q_{th}$ is M) then (GFGrade is VVL) |

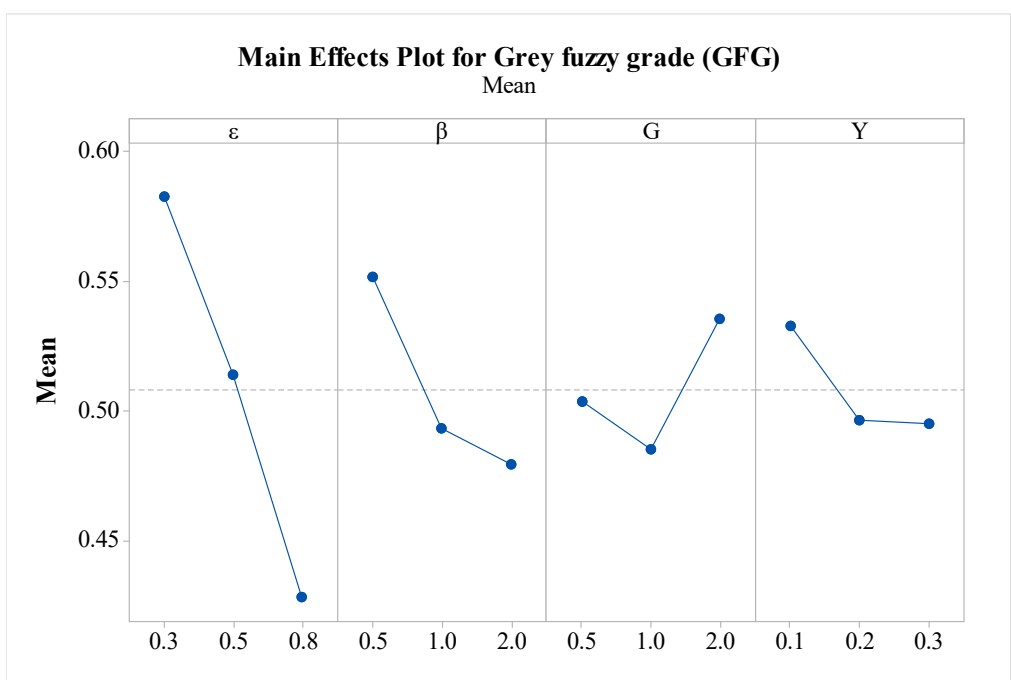

**Figure 8.** Plot for grey-fuzzy grade (GFG).

**Table 10.** Mean grey-fuzzy grade value.

| Input Factors | Levels | | | Delta = (Maximum−Minimum) | Rank of Input Factors | Optimal Levels |
|---|---|---|---|---|---|---|
| | 1st | 2nd | 3rd | | | |
| ε | 0.583 | 0.514 | 0.428 | 0.155 | 1 | 1 |
| β | 0.552 | 0.493 | 0.480 | 0.072 | 2 | 1 |
| G | 0.504 | 0.485 | 0.536 | 0.050 | 3 | 3 |
| y | 0.533 | 0.510 | 0.495 | 0.037 | 4 | 1 |

**Table 11.** Percentage contribution of input terms on GFG by ANOVA.

| Factors | DF | Adj SS | Adj MS | F | P | Contribution (%) |
|---------|-----|----------|----------|-------|-------|------------------|
| $\varepsilon$ | 2 | 0.107927 | 0.053963 | 44.72 | 0.000 | 61.36 |
| $\beta$ | 2 | 0.026438 | 0.013219 | 10.96 | 0.001 | 15.03 |
| G | 2 | 0.011654 | 0.005827 | 4.83 | 0.021 | 6.63 |
| y | 2 | 0.008147 | 0.004073 | 3.38 | 0.057 | 4.62 |
| Residuals | 18 | 0.021719 | 0.001207 | | | 12.36 |
| Sum | 26 | 0.175884 | | | | 100 |

*3.3. Confirmation Test*

To verify the betterment performance at the optimal setting of input terms, a confirmation test was carried out and outputs, such as $W_{is}$, $W_{th}$, $F_{is}$, $F_{th}$, $Q_{is}$, and $Q_{th}$ are 3.684, 2.84, 165.2, 178.3, 5.67, and 6.32, respectively. Furthermore, the obtained result from the confirmation test was expressed by the obtained grey-fuzzy grade ($\gamma_o$). The grey-fuzzy grade for the optimal parametric combination is estimated using Equation (23).

**Table 12.** Optimal and predicted results.

| Measured Responses | Initial Parameters $\varepsilon_3$-$\beta_3$-$G_2$-$y_1$ | Optimal Parameters $\varepsilon_1$-$\beta_1$-$G_3$-$y_1$ | |
|--------------------|----------------------|----------------------|----------|
| | | Prediction | Optimization |
| $W_{is}$ | 17.9991 | 3.917 | 3.684 |
| $W_{th}$ | 6.3785 | 2.932 | 2.840 |
| $F_{is}$ | 17.9991 | 166.243 | 165.200 |
| $F_{th}$ | 6.3785 | 179.341 | 178.300 |
| $Q_{is}$ | 6.4226 | 6.660 | 5.671 |
| $Q_{th}$ | 6.2705 | 6.21 | 6.32 |
| GFG | 0.382 | 0.679 | 0.686 |
| Improvement | | 0.297 | 0.304 |

**4. Conclusions**

This paper investigated the grey-fuzzy hybrid optimization of misaligned rough elliptic bore journal bearing considering misalignment. The obtained results were summarized as follows:

- Rough elliptic bore misaligned journal bearing has a reduced temperature distribution of 20% with less peak pressure; however, it is spread to a more contact area.
- Angular speed and power index have limited influence on bearing dynamic characteristics, except for whirl ratio. The whirl ratio increases with an increase in angular speed. However, at the same angular speed, the whirl ratio shows negligible change with the variation of degree of misalignment.
- The grey-fuzzy hybrid optimization tool provided the optimal level of input parameters, as follows: $\varepsilon_1$ (0.3)-$\beta_1$ (0.5)-$G_3$ (2)-$y_1$ (0.1). At the optimal input parameter condition, the GFG was enhanced by 79.5% from the initial setting value, which ensured the excellent capability of this hybrid tool to optimize misaligned rough elliptic bore journal bearing parameters.
- At the estimated optimal condition, the optimal solutions in terms of $W_{is}$, $W_{th}$, $F_{is}$, $F_{th}$, $Q_{is}$ and $Q_{th}$ are 3.684, 2.84, 165.2, 178.3, 5.67, and 6.32, respectively.

Finally, the limitation of the model is that it is yet to be validated with experimental outcomes. The immediate implications are to validate the numerical output and to develop

an intelligence system to predict the random set of input parameters. This will be applied to journal bearing design using a fuzzy-based optimization technique. If this rule-based optimization technique is tuned with real-life bearing in operation, it can evaluate the optimum performance. The algorithm and methodology developed can be utilized to create intelligence based on neural network.

**Author Contributions:** Conceptualization, S.K.P. and P.C.M.; methodology, S.K.P.; software, P.C.M. and R.K.; validation, S.K.P. and P.C.M.; formal analysis S.K.P.; investigation, S.K.P. and P.C.M.; resources, P.C.M.; data curation, S.K.P.; writing—original draft preparation, S.K.P.; writing—review and editing, P.C.M. and R.K.; visualization, P.C.M.; supervision P.C.M.; project administration, P.C.M. All authors have read and agreed to the published version of the manuscript.

**Funding:** All authors are thankful to MDPI for full waiver consideration of the APC for publication of this manuscript.

**Data Availability Statement:** All data are part of the manuscript. No external data required.

**Conflicts of Interest:** The authors declared no conflict of interest.

**List of Symbols**

| | |
|---|---|
| $A_t / A_l$ | Amplitude of roughness, transverse/longitudinal (μm) |
| $B_{xx}, B_{xz}, B_{zx}, B_{zz}$ | Dimensionless damping coefficient |
| $c$ | Radial clearance (μm) |
| $c_f$ | Specific heat of lubricant J/kgK |
| $\varepsilon$ | Eccentricity ratio (e/c) |
| $h$ | Oil film thickness |
| $K$ | Dimensionless spring constant |
| $M$ | Non-dimensional mass ($mc\omega^2/W$) |
| $p$ | Fluid film pressure (Pa) |
| $p_c$ | Cavitation pressure (Pa) |
| $r_j$ | Journal radius (mm) |
| $t$ | Time (s) |
| $F_\phi$ | Film pressure force, circumferential (N) |
| $F_R$ | Film pressure force, radial (N) |
| $D_m$ | Degree of misalignment |
| $G$ | Non-circularity/ellipticity |
| $F$ | Friction force (N) |
| $N$ | Rotational speed (rpm) |
| $Q$ | Fluid flow |
| $W$ | Load carrying capacity |
| $X_i(x)$ | Normalized data, xth |
| $y$ | Coefficient of roughness |
| $Z(x)$ | Response data |
| $G_i$ | Grey relational grade of $i$th experiment |
| $\mu$ | Lubricant viscosity (ms$^{-1}$) |
| $\mu_o$ | Limiting low shear viscosity (PaS) |
| $\mu_\infty$ | Limiting high shear viscosity (PaS) |
| $\theta$ | Circumferential location (rad) |
| $\theta_c$ | Dimensionless density |
| $\Phi$ | Attitude angle (rad) |
| $\lambda_l$ | Wavelength, longitudinal |
| $\lambda_t$ | Wavelength, transverse |
| $\beta_b$ | Bulk modulus of lubricant |
| $\beta_t$ | Coefficient of thermal expansion, (1/K) |
| $\beta$ | L/D ratio |

| | |
|---|---|
| $\gamma$ | Viscosity-temperature coefficient (1/K) |
| $\rho$ | Density of lubricant (kg/m$^3$) |
| $\rho_0$ | Density of lubricant at ambient temperature (kg/m$^3$) |
| $\nu$ | Poison's ratio |
| $\Omega$ | Whirl ratio |
| $\omega$ | Angular velocity (rad/s) |
| $\omega_p$ | Angular velocity of whirl (rad/s) |

### Nomenclature

| | |
|---|---|
| ANN | Artificial Neural Network |
| ALA | Artificial Life Algorithm |
| DOE | Design of experiment |
| EALA | Extended artificial life algorithm |
| FDM | Finite Difference Method |
| GRA | Grey Relational Analysis |
| GRC | Grey Relational Coefficient |
| GFG | Grey-Fuzzy Grade |
| H | High |
| L | low |
| L/D | Length to diameter ratio |
| M | Medium |
| MF | Membership Function |
| VH | Very High |
| VVH | Very Very High |
| VL | Very Low |
| VVL | Very Very Low |

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
