# Peer review of "Grey-Fuzzy Hybrid Optimization for Thermohydrodynamic Performance Prediction of Misaligned Rough Elliptic Bore Journal Bearing"

_lubricants, doi:10.3390/lubricants10100274_

Round 1

Reviewer 1 Report

The authors theoretically investigated the effects of multiple parameters on the performances of rough elliptic bore misaligned journal bearings. A Grey-fuzzy optimization model was established to predict the performance of imperfect bearings. I think this can be an interesting paper but I do have some serious concerns listed in the following.

Major comments:

1.          After reading chapter 1, I can’t get the point of why the performance of MREBJB needs to be studied. One can easily understand that all imperfect production will affect the performance, but why do we care about the performances of the bearings when it actually happens? Or more specifically, what can we learn from the predicted results? That is the key motivation missing in the manuscript.

2.          Are the parameters listed in Table 2 based on an actual product or just randomly picked numbers? As bearings are widely used all over the world, the parameter used for testing the model should be at least based on real bearings. Please also list the targeted product as a reference.

3.          I think I am somehow lost while reading the manuscript. Is the key contribution of this paper proposing a Grey-fuzzy hybrid optimization model or the model is just a tool and the predicted results are the main contribution? Based on the title, abstract, and chapter 1, it should be the later. But it seems like most (or even all) of the result and discussion section is about the model and there is not even any discussing regarding the predicted results and what we can learn from it. A following question regarding the same issue is that excluding the part that describes the optimization model, the actual “results and discussion” are not even sufficient to form a section (don’t even mention a chapter). The manuscript is highly unbalanced in terms of the content (take chapters 1 and 3 as an example).

Minor comments:

1.          There are way too many typos and grammar errors in the manuscript such as “methodologyto” in line 39. There’s even a missing number presented as “--” in line 229 and a section 2.2 in chapter 3. This really makes me feel the manuscript wasn’t prepare carefully and the authors didn’t even double check before submission. Please carefully double check the whole manuscript again.

2.          The formatting of this paper is off (the misalignment of texts and equations for example).

3.          I think it would be pretty helpful if a schematic of the analyzed bearing is also provided.

4.          Most of the tables provided in the manuscript are too much to be included in the manuscript. It would be much better to have an appendix for those tables.

Author Response

Thank you for giving us the opportunity to revise our manuscript. Please find the response to the review comments attached.

with regards,

Reviewer 2 Report

Dear authors,
The article refers to the usage of fuzzy logic in an optimization problem related to thermohydrodynamic performance prediction.  
A detailed list of main questions and suggestions:
1. You did a literature review, providing a list of 24 references. However, as far as I see, there is no single item from the journal you want to publish. In my opinion, you should better justify your subject as suitable for the journal
2. You should comment more on the results in tables 3-6. For example, I see a discrepancy; in lines 230-231, there is a statement: "The highest value of Kxx of 0.625 occurs at PI of 0.2 and Dm=0.9.". In table 4, row 3 ( PI(0.2), Dm(0.9) ) the Kxx value is 0.34. Is that correct?
3. Can you justify the division of your domains to exactly seven fuzzy sets evenly distributed in the computational domain (lines 302-304, Fig.3,4)?
4. You said you had defined 20 fuzzy rules (lines 305-306). However, you have six input parameters, each divided into seven fuzzy sets. As far as I see, each rule refers to one combination of all input parameters. Therefore, your complete inference system should include 7*7*7*7*7*7 = over 100 000 rules. Can you comment on this?
5. Conclusions should be extended. You just repeated the obtained results. However, you may also formulate some more general conclusions coming from your research.

Editorial and minor issues:
6. Table 2. There is no value of Time constant \Lambda
7. Line 229: "There is --% increase in Kxx" - seems that value is not present
8. You use a strange symbol for celsius degrees (instead of °)
9. Figure 2. In my opinion, it is NOT a fuzzy INTERFERENCE system, but a fuzzy INFERENCE system.

Regards,

Author Response

Thank you for giving us the opportunity to revise our manuscript. Please find the response to the review comments attached here.

with regards,

Round 2

Reviewer 1 Report

I am satisfied with the responses. A minor suggestion: With all those symbols used in the manuscript, I believe a nomenclature will help a lot.

Author Response

The Authors thank the Esteemed Reviewer for giving their valuable time to improve this manuscript, for which the paper attained the standard to be published in MDPI-Lubricant.

Reviewer 2 Report

Dear authors,
I have read the responses to my questions and suggestions carefully. I think the manuscript has been improved sufficiently to be considered for publication.
Best regards,

Author Response

The Authors thank the Esteemed Reviewer for giving his valuable time to improve this manuscript, for which the paper attained the standard to be published in MDPI-Lubricant.